# Towards Robust Blind Face Restoration with Codebook Lookup Transformer

**Shangchen Zhou**    **Kelvin C.K. Chan**    **Chongyi Li**    **Chen Change Loy**

S-Lab, Nanyang Technological University

{s200094, chan0899, chongyi.li, ccloy}@ntu.edu.sg

https://shangchenzhou.com/projects/CodeFormer

## Abstract

Blind face restoration is a highly ill-posed problem that often requires auxiliary guidance to 1) improve the mapping from degraded inputs to desired outputs, or 2) complement high-quality details lost in the inputs. In this paper, we demonstrate that a learned discrete codebook prior in a small proxy space largely reduces the uncertainty and ambiguity of restoration mapping by casting *blind face restoration* as a *code prediction* task, while providing rich visual atoms for generating high-quality faces. Under this paradigm, we propose a Transformer-based prediction network, named *CodeFormer*, to model the global composition and context of the low-quality faces for code prediction, enabling the discovery of natural faces that closely approximate the target faces even when the inputs are severely degraded. To enhance the adaptiveness for different degradation, we also propose a controllable feature transformation module that allows a flexible trade-off between fidelity and quality. Thanks to the expressive codebook prior and global modeling, *CodeFormer* outperforms the state of the arts in both quality and fidelity, showing superior robustness to degradation. Extensive experimental results on synthetic and real-world datasets verify the effectiveness of our method.

## 1   Introduction

Face images captured in the wild often suffer from various degradation, such as compression, blur, and noise. Restoring such images is highly ill-posed as the information loss induced by the degradation leads to infinite plausible high-quality (HQ) outputs given a low-quality (LQ) input. The ill-posedness is further elevated in blind restoration, where the specific degradation is unknown. Despite the progress made with the emergence of deep learning, learning a LQ-HQ mapping without additional guidance in the huge image space is still intractable, leading to the suboptimal restoration quality of earlier approaches. To improve the output quality, auxiliary information that 1) reduces the uncertainty of LQ-HQ mapping and 2) complements high-quality details is indispensable.

Various priors have been used to mitigate the ill-posedness of this problem, including geometric priors [5, 6, 30, 44], reference priors [24–26], and generative priors [2, 37, 43]. Although improved textures and details are observed, these approaches often suffer from *high sensitivity to degradation* or *limited prior expressiveness*. These priors provide insufficient guidance for face restoration, thus their networks essentially resort to the information of LQ input images that are usually highly corrupted. As a result, the LQ-HQ mapping uncertainty still exists, and the output quality is deteriorated by the degradation of the input images. Most recently, based on generative prior, some methods project the degraded faces into a continuous infinite space via iterative latent optimization [27] or direct latent encoding [29]. Despite great realness of outputs, it is difficult to find the accurate latent vectors in case of severe degradation, resulting in low-fidelity results (Fig. 1(d)). To enhance the fidelity, skip connections between encoder and decoder are usually required in this kind of methods [37, 43, 2], as

36th Conference on Neural Information Processing Systems (NeurIPS 2022).

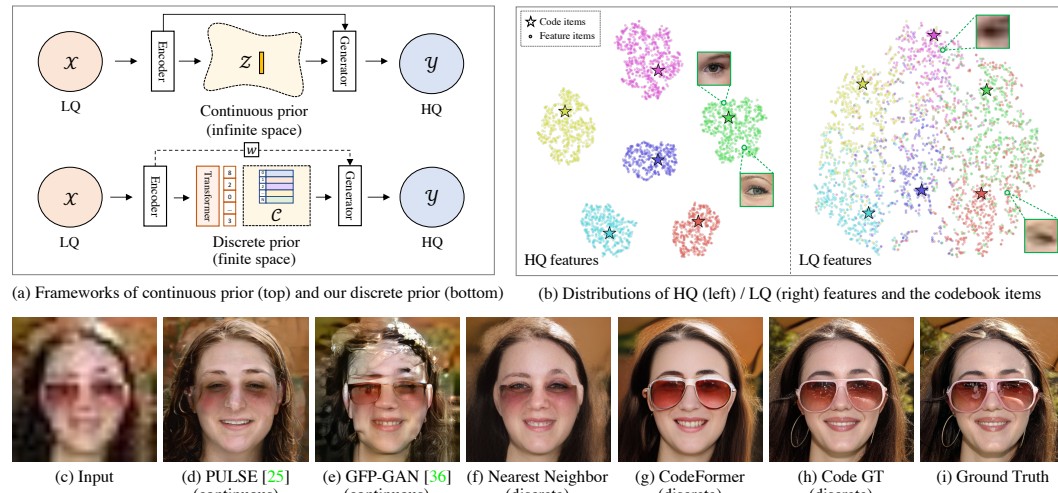

(a) Frameworks of continuous prior (top) and our discrete prior (bottom)

(b) Distributions of HQ (left) / LQ (right) features and the codebook items

(c) Input | (d) PULSE [25] (continuous) | (e) GFP-GAN [36] (continuous) | (f) Nearest Neighbor (discrete) | (g) CodeFormer (discrete) | (h) Code GT (discrete) | (i) Ground Truth

**Figure 1:** An illustration of motivation. (a) Restoration frameworks of continuous generative prior (top) and our discrete codebook prior (bottom). (b) t-SNE [35] visualization for HQ/LQ face features and codebook items. (c) LQ input. (d-e) Results of existing methods with continuous prior (PULSE [27] and GFP-GAN [37]). (f-g) Results of discrete prior (Nearest Neighbor [11, 34] and CodeFormer). (h) Reconstruction results from the code sequence ground truth. (i) HQ ground truth. As shown, (d) PULSE without skip connection shows the low fidelity. (e) GFP-GAN with skip connection alleviates identity issues but introduces notable artifacts. (f) Utilizing nearest neighbor matching for code lookup recovers more accurate facial structure compared with (d-e), but some details such as glasses cannot be restored and some artifacts could be introduced. (g) Employing Transformer for code prediction, our CodeFormer generates best results with both high quality and fidelity.

illustrated in Fig. 1(a) (top), however, such designs meanwhile introduce artifacts in the results when inputs are severely degraded, as shown in Fig. 1(e).

Different from the aforementioned approaches, this work casts *blind face restoration* as a *code prediction* task in a small finite proxy space of the learned *discrete codebook prior*, which shows superior robustness to degradation as well as rich expressiveness. The codebook is learned by self-reconstruction of HQ faces using a vector-quantized autoencoder, which along with decoder stores the rich HQ details for face restoration. In contrast to continuous generative priors [11, 37, 43], the combinations of codebook items form a discrete prior space with only finite cardinality. Through mapping the LQ images to a much smaller proxy space (e.g., 1024 codes), the uncertainty of the LQ-HQ mapping is significantly attenuated, promoting robustness against the diverse degradation, as compared in Figs. 1(d-g). Besides, the codebook space possess greater expressiveness, which perceptually approximates the image space, as shown in Fig. 1(h). This nature allows the network to reduce the reliance on inputs and even be free of skip connections.

Though the discrete representation based on a codebook has been deployed for image generation [4, 11, 34], the accurate code composition for image restoration remains a non-trivial challenge. The existing works look up codebook via nearest-neighbor (NN) feature matching, which is less feasible for image restoration since the intrinsic textures of LQ inputs are usually corrupted. The information loss and diverse degradation in LQ images inevitably distort the feature distribution, prohibiting accurate feature matching. As depicted in Fig. 1(b) (right), even after fine-tuning the encoder on LQ images, the LQ features cannot cluster well to the exact code but spread into other nearby code clusters, thus the nearest-neighbor matching is unreliable in such cases.

Tailored for restoration, we propose a Transformer-based code prediction network, named *Code-Former*, to exploit global compositions and long-range dependencies of LQ faces for better code prediction. Specifically, taking the LQ features as input, the Transformer module predicts the code token sequence which is treated as the discrete representation of the face images in the codebook space. Thanks to the global modeling for remedying the local information loss in LQ images, the proposed *CodeFormer* shows robustness to heavy degradation and keeps overall coherence. Comparing the results presented in Figs. 1(f-g), the proposed *CodeFormer* is able to recover more details than the nearest-neighbor matching, such as the glasses, improving both quality and fidelity of restoration.

Moreover, we propose a *controllable feature transformation* module with an adjustable coefficient to control the information flow from the LQ encoder to decoder. Such design allows a flexible trade-off

between restoration *quality* and *fidelity* so that the continuous image transitions between them can be achieved. This module enhances the adaptiveness of *CodeFormer* under different degradations, e.g., in case of heavy degradation, one could manually reduce the information flow of LQ features carrying degradation to produce high-quality results.

Equipped with the above components, the proposed *CodeFormer* demonstrates superior performance in existing datasets and also our newly introduced *WIDER-Test* dataset that consists of 970 severely degraded faces collected from the WIDER-Face dataset [42]. In addition to face restoration, our method also demonstrates its effectiveness on other challenging tasks such as face inpainting, where long-range clues from other regions are required. Systematic studies and experiments are conducted to demonstrate the merits of our method over previous works.

## 2  Related Work

**Blind Face Restoration.** Since face is highly structured, geometric priors of faces are exploited for blind face restoration. Some methods introduce facial landmarks [6], face parsing maps [5, 30, 41], facial component heatmaps [44], or 3D shapes [16, 28, 48] in their designs. However, such prior information cannot be accurately acquired from degraded faces. Moreover, geometric priors are unable to provide rich details for high-quality face restoration.

Reference-based approaches [9, 24–26] have been proposed to circumvent the above limitations. These methods generally require the references to possess same identity as the input degraded face. For example, Li *et al.* [26] propose a guided face restoration network that consists of a warping subnetwork and a reconstruction subnetwork, and a high-quality guided image of the same identity as input is used for better restoring the facial details. However, such references are not always available. DFDNet [24] pre-constructs dictionaries composed of high-quality facial component features. However, the component-specific dictionary features are still insufficient for high-quality face restoration, especially for the regions out of the dictionary scope (*e.g.*, skin, hair). To alleviate this issue, recent VQGAN-based methods [39, 46] explores a learned HQ dictionary, which contains more generic and rich details face restoration.

Recently, the generative facial priors from pre-trained generators, e.g., StyleGAN2 [21], have been widely explored for blind face restoration. It is utilized via different strategies of iterative latent optimization for effective GAN inversion [12, 27] or direct latent encoding of degraded faces [29]. However, preserving high fidelity of the restored faces is challenging when one projects the degraded faces into the continuous infinite latent space. To relieve this issue, GLEAN [2, 3], GPEN [43], and GFPGAN [37] embed the generative prior into encoder-decoder network structures, with additional structural information from input images as guidance. Despite the improvement of fidelity, these methods highly rely on inputs through the skip connections, which could introduce artifacts to results when inputs are severely corrupted.

**Dictionary Learning.** Sparse representation with learned dictionaries has demonstrated its superiority in image restoration tasks, such as super-resolution [13, 32, 33, 40] and denoising [10]. However, these methods usually require an iterative optimization to learn the dictionaries and sparse coding, suffering from high computational cost. Despite the inefficiency, their high-level insight into exploring a HQ dictionary has inspired reference-based restoration networks, e.g., LUT [18] and self-reference [47], as well as synthesis methods [11, 34]. Jo and Kim [18] construct a look-up table (LUT) by transferring the network output values to a LUT, so that only a simple value retrieval is needed during inference. However, storing HQ textures in the image domain usually requires a huge LUT, limiting its practicality. VQVAE [34] is first to introduce a highly compressed codebook learned by a vector-quantized autoencoder model. VQGAN [11] further adopts the adversarial loss and perceptual loss to enhance perceptual quality at a high compression rate, significantly reducing the codebook size without sacrificing its expressiveness. Unlike the large hand-crafted dictionary [18, 24], the learnable codebook automatically learns optimal elements for HQ image reconstruction, providing superior efficiency and expressiveness as well as circumventing the laborious dictionary design. Inspired by the codebook learning, this paper investigates a discrete proxy space for blind face restoration. Different from recent VQGAN-based approaches [39, 46], we exploit the discrete codebook prior by predicting code sequences via global modeling, and we secure prior effectiveness by fixing the encoder. Such designs allow our method to take full advantage of the codebook so that it does not depend on the feature fusion with LQ cues, significantly enhancing the robustness of face restoration.

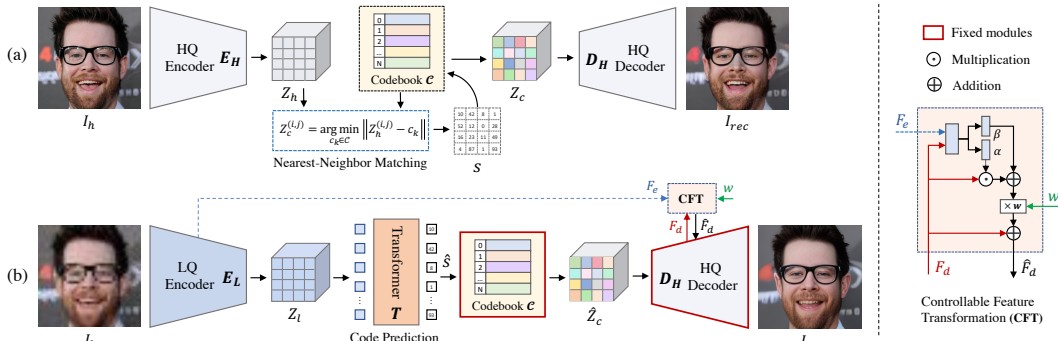

**Figure 2:** Framework of CodeFormer. (a) We first learn a discrete codebook and a decoder to store high-quality visual parts of face images via self-reconstruction learning. (b) With fixed codebook and decoder, we then introduce a Transformer module for code sequence prediction, modeling the global face composition of low-quality inputs. Besides, a controllable feature transformation module is used to control the information flow from LQ encoder to decoder. Note that this connection is optional, which can be disabled to avoid adverse effects when inputs are severely degraded, and one can adjust a scalar weight $w$ to trade between *quality* and *fidelity*.

# 3  Methodology

The main focus of this work is to exploit a discrete representation space that reduces the uncertainty of restoration mapping and complements high-quality details for the degraded inputs. Since local textures and details are lost and corrupted in low-quality inputs, we employ a Transformer module to model the global composition of natural faces, which remedies the local information loss, enabling high-quality restoration. The overall framework is illustrated in Fig. 2.

We first incorporate the idea of vector quantization [11, 34] and pre-train a quantized autoencoder through self-reconstruction to obtain a discrete codebook and the corresponding decoder (Sec. 3.1). The prior from the codebook combination and decoder is then used for face restoration. Based on this codebook prior, we then employ a Transformer for accurate prediction of code combination from the low-quality inputs (Sec. 3.2). In addition, a controllable feature transformation module is introduced to exploit a flexible trade-off between restoration quality and fidelity (Sec. 3.3). The training of our method is divided into three stages accordingly.

## 3.1  Codebook Learning (Stage I)

To reduce uncertainty of the LQ-HQ mapping and complement high-quality details for restoration, we first pre-train the quantized autoencoder to learn a context-rich codebook, which improves network expressiveness as well as robustness against degradation.

As shown in Fig. 2(a), the HQ face image $I_h \in \mathbb{R}^{H \times W \times 3}$ is first embedded as a compressed feature $Z_h \in \mathbb{R}^{m \times n \times d}$ by an encoder $E_H$. Following VQVAE [34] and VQGAN [11], we replace each "pixel" in $Z_h$ with the nearest item in the learnable codebook $\mathcal{C} = \{c_k \in \mathbb{R}^d\}_{k=0}^N$ to obtain the quantized feature $Z_c \in \mathbb{R}^{m \times n \times d}$ and the corresponding code token sequence $s \in \{0, \cdots, N-1\}^{m \cdot n}$:

$$Z_c^{(i,j)} = \arg\min_{c_k \in \mathcal{C}} \|Z_h^{(i,j)} - c_k\|_2; \quad s^{(i,j)} = \arg\min_k \|Z_h^{(i,j)} - c_k\|_2. \tag{1}$$

The decoder $D_H$ then reconstructs the high-quality face image $I_{rec}$ given $Z_c$. The $m \cdot n$ code token sequence $s$ forms a new latent discrete representation that specifies the respective code index of the learned codebook, i.e., $Z_c^{(i,j)} = c_k$ when $s^{(i,j)} = k$.

**Training Objectives.** To train the quantized autoencoder with a codebook, we adopt three image-level reconstruction losses: L1 loss $\mathcal{L}_1$, perceptual loss [19, 45] $\mathcal{L}_{per}$, and adversarial loss [11] $\mathcal{L}_{adv}$:

$$\mathcal{L}_1 = \|I_h - I_{rec}\|_1; \quad \mathcal{L}_{per} = \|\Phi(I_h) - \Phi(I_{rec})\|_2^2; \quad \mathcal{L}_{adv} = [\log D(I_h) + \log(1 - D(I_{rec}))], \tag{2}$$

where $\Phi$ denotes the feature extractor of VGG19 [31]. Since, image-level losses are underconstrained when updating the codebook items, we also adopt the intermediate code-level loss [11, 34] $\mathcal{L}_{code}^{feat}$ to reduce the distance between codebook $\mathcal{C}$ and input feature embeddings $Z_h$:

$$\mathcal{L}_{code}^{feat} = \|\text{sg}(Z_h) - Z_c\|_2^2 + \beta\|Z_h - \text{sg}(Z_c)\|_2^2, \tag{3}$$

where sg$(\cdot)$ stands for the stop-gradient operator and $\beta = 0.25$ is a weight trade-off for the update rates of the encoder and codebook. Since the quantization operation in Eq. (1) is non-differentiable, we adopt straight-through gradient estimator [11, 34] to copy the gradients from the decoder to the encoder. The complete objective of codebook prior learning $\mathcal{L}_{codebook}$ is:

$$\mathcal{L}_{codebook} = \mathcal{L}_1 + \mathcal{L}_{per} + \mathcal{L}_{code}^{feat} + \lambda_{adv} \cdot \mathcal{L}_{adv}, \tag{4}$$

where $\lambda_{adv}$ is set to 0.8 in our experiments.

**Codebook Settings.** Our encoder $E_H$ and decoder $D_H$ consist of 12 residual blocks and 5 resize layers for downsampling and upsampling, respectively. Hence we obtain a large compression ratio of $r = H/n = W/m = 32$, which leads to a great robustness against degradation and a tractable computational cost for our global modeling in Stage II. Although more codebook items could ease reconstruction, the redundant elements could cause ambiguity in subsequent code predictions. Hence, we set the item number $N$ of codebook to 1024, which is sufficient for accurate face reconstruction. Besides, the code dimension $d$ is set to 256.

## 3.2 Codebook Lookup Transformer Learning (Stage II)

Due to corruptions of textures in LQ faces, the nearest-neighbour (NN) matching in Eq. (1) usually fails in finding accurate codes for face restoration. As depicted in Fig. 1(b), LQ features with diverse degradation could deviate from the correct code and be grouped into nearby clusters, resulting in undesirable restoration results, as shown in Fig. 1(f). To mitigate the problem, we employ a Transformer to model global interrelations for better code prediction.

Built upon the learned autoencoder presented in Sec. 3.1, as shown in Fig. 2(b), we insert a Transformer [36] module containing nine self-attention blocks following the encoder. We fix the codebook $\mathcal{C}$ and decoder $D_H$ and finetune the encoder $E_H$ for restoration. The finetuned encoder is denoted as $E_L$. To obtain the LQ features $Z_l \in \mathbb{R}^{m \times n \times d}$ through $E_L$, we first unfold the features into $m \cdot n$ vectors $Z_l^v \in \mathbb{R}^{(m \cdot n) \times d}$, and then feed them to the Transformer module. The $s$-th self-attention block of Transformer computes as the following:

$$X_{s+1} = \text{Softmax}(Q_s K_s)V_s + X_s, \tag{5}$$

where $X_0 = Z_l^v$. The query $Q$, key $K$, and value $V$ are obtained from $X_s$ through linear layers. We add a sinusoidal positional embedding $\mathcal{P} \in \mathbb{R}^{(m \cdot n) \times d}$ [1, 7] on the queries $Q$ and the keys $K$ to increase the expressiveness of modeling sequential representation. Following the self-attention blocks, a Linear layer is adopted to project features to the dimension of $(m \cdot n) \times N$. Overall, taking the encoding feature $Z_l^v$ as an input, the Transformer predicts the $m \cdot n$ code sequence $\hat{s} \in \{0, \cdots, |N| - 1\}^{m \cdot n}$ in form of the probability of the $N$ code items. The predicted code sequence then retrieves the $m \cdot n$ respective code items from the learned codebook, forming the quantized feature $\hat{Z}_c \in \mathbb{R}^{m \times n \times d}$ that produces a high-quality face image through the fixed decoder $D_H$. Thanks to our large compression ratio (*i.e.*, 32), our Transformer is effective and efficient in modeling global correlations of LQ face images.

**Training Objectives.** We train Transformer module $T$ as well as finetune the encoder $E_L$ for restoration, while the codebook $\mathcal{C}$ and decoder $D_H$ are kept fixed. Instead of employing reconstruction loss and adversarial loss in the image-level, only code-level losses are required in this stage: 1) cross-entropy loss $\mathcal{L}_{code}^{token}$ for code token prediction supervision, and 2) L2 loss $\mathcal{L}_{code}^{feat'}$ to force the LQ feature $Z_l$ to approach the quantized feature $Z_c$ from codebook, which eases the difficulty of token prediction learning:

$$\mathcal{L}_{code}^{token} = \sum_{i=0}^{mn-1} -s_i \log(\hat{s}_i); \quad \mathcal{L}_{code}^{feat'} = \|Z_l - \text{sg}(Z_c)\|_2^2, \tag{6}$$

where the ground truth of latent code $s$ is obtained from the pre-trained autoencoder in Stage I (Sec. 3.1), thus the quantized feature $Z_c$ is then retrieved from codebook according to the $s$. The final objective of Transformer learning is:

$$\mathcal{L}_{tf} = \lambda_{token} \cdot \mathcal{L}_{code}^{token} + \mathcal{L}_{code}^{feat'}, \tag{7}$$

where $\lambda_{token}$ is set to 0.5 in our method. Note that our network after this stage has already equipped with superior robustness and effectiveness in face restoration, especially for severely degraded faces.

### 3.3 Controllable Feature Transformation (Stage III)

Despite our Stage II has obtained a great face restoration model, we also investigate a flexible tradeoff between *quality* and *fidelity* of face restoration. Thus, we propose the controllable feature transformation (CFT) module to control information flow from LQ encoder $E_L$ to decoder $D_H$. Specifically, as shown in Fig. 2, the LQ features $F_e$ are used to slightly tune the decoder features $F_d$ through spatial feature transformation [38] with the affine parameters of $\alpha$ and $\beta$. An adjustable coefficient $w \in [0, 1]$ is then used to control the relative importance of the inputs:

$$\hat{F}_d = F_d + (\alpha \odot F_d + \beta) \times w; \quad \alpha, \beta = \mathcal{P}_\theta(c(F_d, F_e)), \tag{8}$$

where $\mathcal{P}_\theta$ denotes a stack of convolutions that predicts $\alpha$ and $\beta$ from the concatenated features of $c(F_e, F_d)$. We adopt the CFT modules at multiple scales $s \in \{32, 64, 128, 256\}$ between encoder and decoder. Such a design allows our network to remain high fidelity for mild degradation and high quality for heavy degradation. Specifically, one could use a small $w$ to reduce the reliance on input LQ images with heavy degradation, thus producing high-quality outputs. The larger $w$ will introduce more information from LQ images to enhance the fidelity in case of mild degradation.

**Training Objectives.** To train the controllable modules and finetune the encoder $E_L$ in this stage, we keep the code-level losses of $\mathcal{L}_{tf}$ in Stage II, and also add image-level losses of $\mathcal{L}_1$, $\mathcal{L}_{per}$, and $\mathcal{L}_{adv}$, which are the same as that in Stage I except that $I_{rec}$ is replaced by restoration output $I_{res}$. The complete loss for this stage is the sum of above losses weighted with their original weight factors. We set the $w$ to 1 during training of this stage, which then allows network to achieve continuous transitions of results by adjusting $w$ in $[0, 1]$ during inference. For inference, unless otherwise stated, we set the $w = 0.5$ by default to make a good balance between the quality and fidelity of outputs.

## 4 Experiments

### 4.1 Datasets

**Training Dataset.** We train our models on the FFHQ dataset [21], which contains 70,000 high-quality (HQ) images, and all images are resized to 512×512 for training. To form training pairs, we synthesize LQ images $I_l$ from the HQ counterparts $I_h$ with the following degradation model [24, 37, 43]:

$$I_l = \{[(I_h \otimes k_\sigma)_{\downarrow r} + n_\delta]_{\text{JPEG}_q}\}_{\uparrow r}, \tag{9}$$

where the HQ image $I_h$ is first convolved with a Gaussian kernel $k_\sigma$, followed by a downsampling of scale $r$. After that, additive Gaussian noise $n_\delta$ is added to the images, and then JPEG compression with quality factor $q$ is applied. Finally, the LQ image is resized back to 512×512. We randomly sample $\sigma$, $r$, $\delta$, and $q$ from $[1, 15]$, $[1, 30]$, $[0, 20]$, and $[30, 90]$, respectively.

**Testing Datasets.** We evaluate our method on a synthetic dataset CelebA-Test and three real-world datasets: LFW-Test, WebPhoto-Test, and our proposed WIDER-Test. CelebA-Test contains 3,000 images selected from the CelebA-HQ dataset [20], where LQ images are synthesized under the same degradation range as our training settings. The three real-world datasets respectively contain three different degrees of degradation, *i.e.*, mild (LFW-Test), medium (WebPhoto-Test), and heavy (WIDER-Test). LFW-Test consists of the first image of each person in LFW dataset [17], containing 1,711 images. WebPhoto-Test [37] consists of 407 low-quality faces collected from the Internet. Our WIDER-Test comprises 970 severely degraded face images from the WIDER Face dataset [42], providing a more challenging dataset to evaluate the generalizability and robustness of blind face restoration methods.

### 4.2 Experimental Settings and Metrics

**Settings.** We represent a face image of $512 \times 512 \times 3$ as a $16 \times 16$ code sequence. For all stages of training, we use the Adam [23] optimizer with a batch size of 16. We set the learning rate to $8 \times 10^{-5}$ for stages I and II, and adopt a smaller learning rate of $2 \times 10^{-5}$ for stage III. The three stages are trained with 1.5M, 200K, and 20K iterations, respectively. Our method is implemented with the PyTorch framework and trained using four NVIDIA Tesla V100 GPUs.

**Metrics.** For the evaluation on CelebA-Test with ground truth, we adopt PSNR, SSIM, and LPIPS [45] as metrics. We also evaluate the identity preservation using the cosine similarity of features from

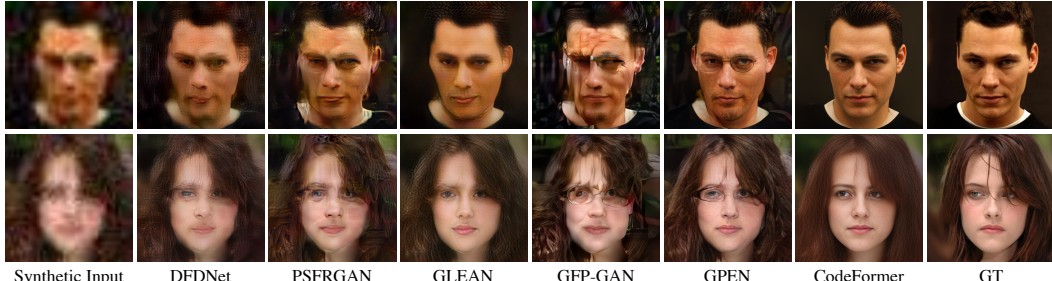

| Synthetic Input | DFDNet | PSFRGAN | GLEAN | GFP-GAN | GPEN | CodeFormer | GT |

**Figure 3:** Qualitative comparison on the CelebA-Test. Even though input faces are severely degraded, our CodeFormer produces high-quality faces with faithful details. (**Zoom in for details**)

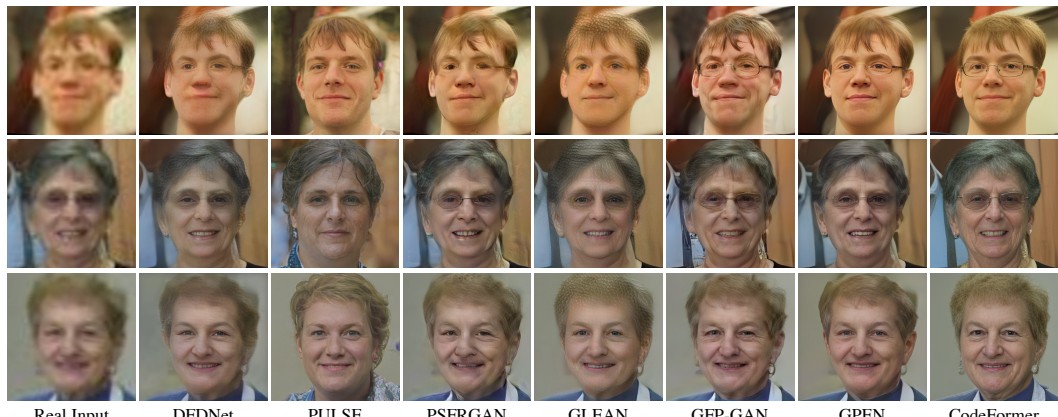

| Real Input | DFDNet | PULSE | PSFRGAN | GLEAN | GFP-GAN | GPEN | CodeFormer |

**Figure 4:** Qualitative comparison on real-world faces. Our CodeFormer is able to restore high-quality faces, showing robustness to the heavy degradation. (**Zoom in for details**)

ArcFace network [8], denoted as IDS. For the evaluation on real-world datasets without ground truth, we employ the widely-used non-reference perceptual metrics: FID [15] and MUSIQ (KonIQ) [22].

## 4.3 Comparisons with State-of-the-Art Methods

We compare the proposed CodeFormer with state-of-the-art methods, including PULSE [27], DFD-Net [24], PSFRGAN [5], GLEAN [3], GFP-GAN [37], and GPEN [43]. We conduct extensive comparisons on both synthetic and real-world datasets.

**Evaluation on Synthetic Dataset.** We first show the quantitative comparison on the CelebA-Test in Table 1. In terms of the image quality metrics LPIPS, FID, and MUSIQ, our CodeFormer achieves the best scores than existing methods. Besides, it also faithfully preserves the identity, reflected by the highest IDS score and PSNR. Additionally, we present the qualitative comparison in Fig. 3. The compared methods fail to produce pleasant restoration results, e.g., DFDNet [24], PSFRGAN [5], GFP-GAN [37], and GPEN [43] introduce obvious artifacts and GLEAN [3] produces over-smoothed

**Table 1:** Quantitative comparison on the **CelebA-Test**. Red and blue indicate the best and the second best performance, respectively. The result of Code GT is the upper bound of CodeFormer.

| Methods | LPIPS↓ | FID↓ | MUSIQ↑ | IDS↑ | PSNR↑ | SSIM↑ |
|---|---|---|---|---|---|---|
| Input | 0.712 | 295.73 | 15.16 | 0.32 | 21.53 | 0.623 |
| PULSE [27] | 0.406 | 72.94 | 67.39 | 0.30 | 21.38 | 0.571 |
| DFDNet [24] | 0.466 | 85.15 | 57.00 | 0.42 | 21.24 | 0.562 |
| PSFRGAN [5] | 0.395 | 62.05 | 65.93 | 0.43 | 20.91 | 0.549 |
| GLEAN [3] | 0.371 | 59.87 | 61.59 | 0.51 | 21.59 | 0.598 |
| GFP-GAN [37] | 0.391 | 58.36 | 67.84 | 0.42 | 20.37 | 0.545 |
| GPEN [43] | 0.349 | 59.70 | 71.53 | 0.54 | 21.26 | 0.565 |
| **CodeFormer (ours)** | 0.299 | 60.62 | 73.79 | 0.60 | 22.18 | 0.610 |
| Code GT | 0.124 | 54.31 | 71.94* | 0.89 | 25.43 | 0.749 |
| GT | 0 | 51.40 | 72.02* | 1 | ∞ | 1 |

**Table 2:** Quantitative comparison on the *real-world* **LFW-Test**, **WebPhoto-Test**, and **WIDER-Test**. Red and blue indicate the best and the second best performance, respectively.

| Dataset | LFW-Test | | WebPhoto-Test | | WIDER-Test | |
|---|---|---|---|---|---|---|
| Degradation | mild | | medium | | heavy | |
| Methods | FID↓ | MUSIQ↑ | FID↓ | MUSIQ↑ | FID↓ | MUSIQ↑ |
| Input | 137.56 | 25.05 | 170.11 | 19.24 | 202.06 | 15.57 |
| PULSE [27] | 64.86 | 66.98 | 86.45 | 66.57 | 73.59 | 65.36 |
| DFDNet [24] | 62.57 | 67.95 | 100.68 | 63.81 | 57.84 | 59.34 |
| PSFRGAN [5] | 51.89 | 69.21 | 88.45 | 67.09 | 51.16 | 67.27 |
| GLEAN [3] | 53.49 | 66.48 | 105.63 | 61.30 | 47.11 | 60.68 |
| GFP-GAN [37] | 49.96 | 68.95 | 87.35 | 68.04 | 40.59 | 68.26 |
| GPEN [43] | 57.58 | 73.59 | 81.77 | 73.41 | 46.99 | 72.36 |
| **CodeFormer (ours)** | 52.02 | 71.43 | 78.87 | 70.51 | 39.06 | 69.31 |

| Exp. | Networks | | | Code Lookup | | Metrics | |
|---|---|---|---|---|---|---|---|
| | Codebook | Transformer | Fix Decoder | NN | Code Pred. | LPIPS↓ | IDS↑ |
| (a) | | | ✓ | | | 0.420 | 0.47 |
| (b) | ✓ | | ✓ | ✓ | | 0.397 | 0.51 |
| (c) | ✓ | | ✓ | | ✓ | 0.351 | 0.55 |
| (e) | ✓ | ✓ | | | ✓ | 0.379 | 0.52 |
| (f) (ours, w=1) | ✓ | ✓ | ✓ | | ✓ | **0.297** | **0.60** |
| (g) (ours, w=0) | ✓ | ✓ | ✓ | | ✓ | 0.307 | 0.58 |

**Table 3:** Ablation studies of variant networks and code lookup methods on the CelebA-Test. Removing 'Codebook' means the network is a general encoder-decoder structure. '$w$' is an adjustable coefficient of CFT modules that controls the information flow from encoder.

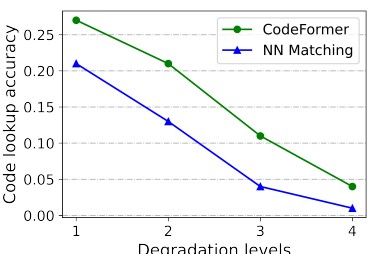

**Figure 6:** Curve comparison on code sequence prediction accuracy.

results that lack facial details. Moreover, all compared methods are unable to preserve the identity. Thanks to the expressive codebook prior and global modeling, CodeFormer not only produces high-quality faces but also preserves the identity well, even when inputs are highly degraded.

**Evaluation on Real-world Datasets.** As presented in Table 2, our CodeFormer achieves comparable perceptual quality of FID score with the compared methods on the real-world testing datasets with mild and medium degradation, and the best score on the testing dataset with heavy degradation. Although PULSE [27] also obtains good perceptual MUSIQ score, it cannot preserve the identity of input images, as the identity score of IDS and visual results respectively suggested in Table 1 and Fig. 4. From the visual comparisons in Fig. 4, it is observed that our method shows exceptional robustness to the real heavy degradation and produces most visually pleasing results. Notably, CodeFormer successfully preserves the identity and produces natural results with rich details.

## 4.4 Ablation Studies

**Effectiveness of Codebook Space.** We first investigate the effectiveness of the codebook space. As shown in Exp. (a) of Table 3, removing the codebook (*i.e.*, directly feeding the encoder features $Z_l$ to the decoder) results in worse LPIPS and IDS scores. The results suggest that the discrete space of codebook is the key to ensure the robustness and effectiveness of our model.

**Superiority of Transformer-based Code Prediction.** To verify the superiority of our Transformer-based code prediction for codebook lookup, we compare it with two different solutions, *i.e.*, nearest-neighbour (NN) matching, *i.e.*, Exp. (b), and a CNN-based code prediction module, *i.e.*, Exp. (c), that adopts a Linear layer for prediction following encoder $E_L$. As shown in Table 3, the comparison of Exps. (b) and (c) indicates that adopting code prediction for codebook lookup is more effective than NN feature matching. However, the local nature of convolution operation of CNNs restricts its modeling capability for long code sequence prediction. In comparison to the pure CNN-based method, *i.e.*, Exp. (c), our Transformer-based solution produces better-fidelity results in terms of LPIPS and IDS scores, as well as higher accuracy of code prediction under all degradation degrees, as shown in Fig. 6. Besides, the superiority of CodeFormer is also demonstrated in visual comparisons shown in Fig. 5 and Fig. 9.

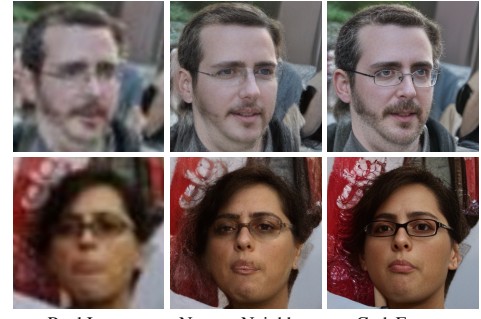

Real Input    Nearest Neighbor    CodeFormer

**Figure 5:** Qualitative comparisons of different codebook lookup methods.

**Importance of Fixed Decoder.** Unlike the large dictionary ($\sim$3.2G) in DFDNet [24], which aims to store a massive quantity of facial details, we deliberately adopt a compact codebook $\mathcal{C} \in \mathbb{R}^{N \times d}$ with $N{=}1024$ and $d{=}256$ that only keeps the essential codes for face restoration, which then activate the detailed cues stored in the pre-trained decoder. Thus, the codebook must be used alongside the decoder to fully unleash its potential. To vindicate our design, we conduct two studies: 1) fixing both codebook and decoder, *i.e.*, Exp. (g), and 2) fixing codebook but fine-tuning decoder, *i.e.*, Exp. (e). Table 3 shows fine-tuning decoder deteriorates the performance, validating our statement. This is because fine-tuning the decoder destroys the learned prior that is held by the pre-trained codebook and decoder, resulting in suboptimal performance. Therefore, we keep the decoder fixed in our method.

higher quality ← → higher fidelity

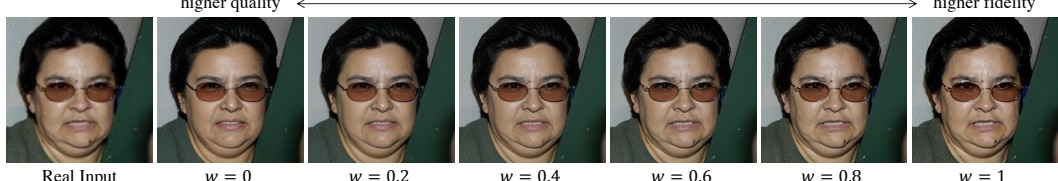

| Real Input | $w = 0$ | $w = 0.2$ | $w = 0.4$ | $w = 0.6$ | $w = 0.8$ | $w = 1$ |

**Figure 7:** CFT module is capable to generate continuous transitions between image quality and fidelity.

**Flexibility of Controllable Feature Transformation Module.** Considering the diverse degradation in real-world LQ face images, we provide a controllable feature transformation module (CFT) to allow a flexible trade-off between quality and fidelity. As shown in Fig. 7, a smaller $w$ tends to produce a high-quality result while a larger $w$ improves the fidelity. While such a flexibility is rarely explored in previous work, here we show that it is an appealing strategy to improves the adaptiveness of our method for different scenarios. As shown in Table 3, Exp. (f), *i.e.*, setting the coefficient $w$ to 1 increases the reconstruction and identity scores but decreases the visual quality. In this work, we trade between the quality and fidelity, and set the coefficient $w$ to 0.5 by default.

## 4.5 Running time

We compare the running time of state-of-the-art methods [27, 24, 5, 2, 37, 43] and the proposed CodeFormer. All existing methods are evaluated on $512^2$ face images using their publicly available code. As shown in Table 5, the proposed CodeFormer has a similar running time as PSFRGAN [5] and GPEN [43] that can infer one image within 0.1s. Meanwhile, our method achieves the best performance in terms of LPIPS on the Celeb-Test dataset.

**Table 5:** Running time of different networks. All methods are evaluated on $512^2$ input images using an NVIDIA Tesla V100 GPU. '__' indicates the running time is less than 0.1s per test image.

|  | PULSE [27] | DFDNet [24] | PSFRGAN [5] | GLEAN [2] | GFP-GAN [37] | GPEN [43] | **CodeFormer (Ours)** |
|---|---|---|---|---|---|---|---|
| Time (sec) | 48.955 | 0.179 | 0.065 | 0.132 | 0.126 | **0.055** | 0.070 |
| LPIPS↓ | 0.406 | 0.466 | 0.395 | 0.371 | 0.391 | 0.349 | **0.299** |

## 4.6 Extensions

**Face Color Enhancement.** We finetune our model on face color enhancement using the same color augmentations (random color jitter and grayscale conversion) as GFP-GAN (v1) [37]. We compare our method with GFP-GAN (v1) [37] on the real-world old photos (from CelebChild-Test dataset [37]) with color loss. The proposed CodeFormer generates high-quality face images with more natural color and faithful details.

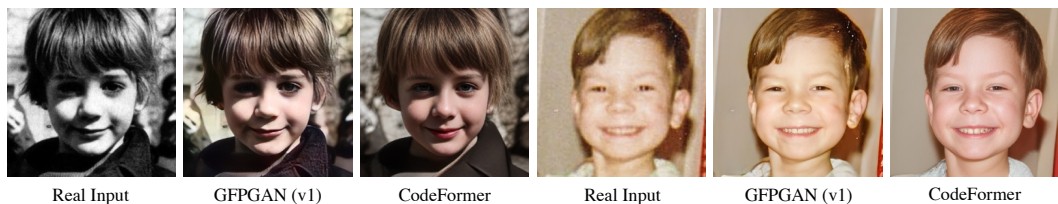

| Real Input | GFPGAN (v1) | CodeFormer | Real Input | GFPGAN (v1) | CodeFormer |

**Figure 8:** Visual comparison of face color enhancement on the real-world old face photos.

**Face Inpainting.** The proposed Codeformer can be easily extended to face inpainting, and it shows great performance even in large mask ratios. To build training pairs, we use a publicly available script [43] to randomly draw irregular polyline masks for generating masked faces. We compare our method with two state-of-the-art face inpainting methods CTSDG [14] and GPEN [43], as well as Nearest-Neighbor matching for codebook lookup. As shown in Fig. 9, CTSDG and GPEN struggle in cases with large masks. Using Nearest-Neighbor matching within our framework roughly reconstructs the face structures, but it also fails in restoring complete visual parts such as the glasses and the eyes. In contrast, our CodeFormer generates high-quality natural faces without strokes and artifacts.

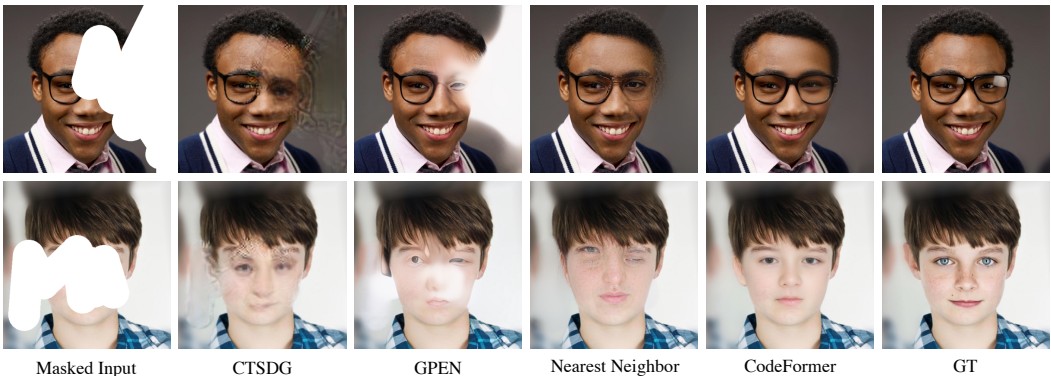

| Masked Input | CTSDG | GPEN | Nearest Neighbor | CodeFormer | GT |

**Figure 9:** Visual comparison with state-of-the-art face inpainting methods on the challenging cases.

### 4.7 Limitation

Our method is built on a pre-trained autoencoder with a codebook. Thus, the capability and expressiveness of the autoencoder could affect the performance of our method. 1) Though the identity inconsistency issue is significantly relieved by the Transformer's global modeling, inconsistency still exists in some rare visual parts such as accessories, where the current codebook space cannot seamlessly represent the image space. Using multiple scales in the codebook space to explore more fine-grained visual quantization may be a solution. 2) Although CodeFormer exhibits great robustness in most cases, when it comes to side faces, CodeFormer offers limited superiority to other methods and also cannot produce good results, as failure cases shown in Fig. 10. This is expected because there are only few side faces in the FFHQ training dataset, thus, the codebook is unable to learn sufficient codes for this case, leading to less effectiveness in both reconstruction and restoration.

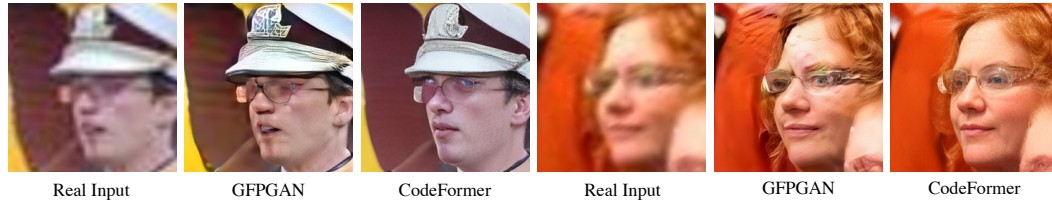

| Real Input | GFPGAN | CodeFormer | Real Input | GFPGAN | CodeFormer |

**Figure 10:** Failure cases tend to occur on side faces, which could be caused by the limited number of side face images in the training dataset of FFHQ.

## 5 Conclusion

This paper aims to address the fundamental challenges in blind face restoration. With a learned small discrete but expressive codebook space, we turn face restoration to code token prediction, significantly reducing the uncertainty of restoration mapping and easing the learning of restoration network. To remedy the local loss, we explore global composition and dependency from degraded faces via an expressive Transformer module for better code prediction. Benefiting from these designs, our method shows great expressiveness and strong robustness against heavy degradation. To enhance the adaptiveness of our method for different degradation, we also propose a controllable feature transformation module that allows a flexible trade-off between fidelity and quality. Experimental results suggest the superiority and effectiveness of our method.

### Acknowledgement

This study is supported under the RIE2020 Industry Alignment Fund – Industry Collaboration Projects (IAF-ICP) Funding Initiative, as well as cash and in-kind contribution from the industry partner(s). It is also partially supported by the NTU NAP grant.

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
