# OpenReview forum: "Towards Robust Blind Face Restoration with Codebook Lookup Transformer"
_NeurIPS.cc/2022/Conference — NeurIPS 2022 Accept_

### Official Review · Reviewer_ozVV · 2022-07-05

**Rating:** 7
**Confidence:** 5
**Soundness:** 4 excellent
**Presentation:** 4 excellent
**Contribution:** 3 good

**Summary:**

This paper deals with blind face restoration in a learned small codebook space. Using a Transformer to model the global composition of LQ faces, the original restoration task is cast into a code prediction task, which is more robust to degradation. To balance the tradeoff between the quality and fidelity of the results, this work proposes to introduce the controllable modules through the connections between encoder and decoder. The proposed CodeFormer outperforms previous methods on multiple datasets and tasks.

**Questions:**

1. The proposed network utilizes a learned codebook as a dictionary, which is similar with the design of DFDNet. As far as I know, DFDNet runs very slowly due to the time-consuming feature matching with its multi-scale dictionaries. Additionally, the proposed method also adopts a Transformer module. How about its runtime?

2. I wonder why the authors train the Stage II and III separately? Is it possible to train them jointly to obtain the final model?

3. This paper adopts a large compression rate of 32 when learn the codebook and autoencoder. However the smaller compression rate, such as 16, could have greater reconstruction capability, which may improve the fidelity of outputs. The discussions or study on the choosing of compression rate could be interesting.

Minor typos:
- In L100, ‘$\alpha$’ -> ‘$\theta$’
- In Eq. (6), should $Z_h$ be $Z_l$?

**Strengths And Weaknesses:**

Strengths:
1. This paper is well organized and nicely written. The presentation of motivation is clear and smooth.

2. Their practice of transforming the image restoration into a code prediction problem is reasonable.

3. The experiments and study are complete and comprehensive. It is also good to provide an interesting video demo. Besides the blind face restoration, the authors also provide an extension of the proposed idea to other tasks such as image inpainting and colorization. The ablation studies are also sufficient to show the efficacy of key components.

4. The visual results are promising, even on heavy degradations.

Weaknesses:

The learned codebook has been investigated for image generation, as I understand, it is hard to perfectly reconstruct the original HQ faces, which will lead lower fidelity of results when applied to restoration. It would be helpful to provide some concise discussions about the reconstruction capability of codebook and any potential fidelity issues.

---

> ### Author Response · Authors · 2022-08-02
> **Response to Assigned Reviewer ozVV**
>
> Thank you for the positive and insightful comments on the novelty and performance of the proposed method, such as well-organized paper, reasonable design, good writing, promising results, and comprehensive study. The raised concerns are addressed as follows.
>
> **[Q1] Provide some concise discussions about the reconstruction capability of codebook and any potential fidelity issues.**
>
> The reconstruction capability of codebook is affected by the number of code items. We conducted an ablation study to explore their relationships. As shown in Table 1 of the supplementary, the reconstruction performance (LPIPS and PSNR) is better when more codebook items are activated and learned. To maintain the high fidelity of the reconstructed faces from codebook, we adopt a larger codebook with 1024 items. We also show that a 1024 codebook is sufficient to encode facial details, while more code items over 1024 cannot significantly enhance the reconstruction capability. Please check out [Q1] of Reviewer pfoY.
>
> Besides, we introduced the controllable feature transformation modules to control the information flow from the encoder to decoder, which also helps to complement code composition errors and expression defects. High-fidelity results can be obtained by setting the controllable coefficient *w* to 1.
>
>
> **[Q2] Runtime comparison.**
>
> We provided the runtime comparison in Table 3 of the supplementary. The proposed CodeFormer has a similar runtime with PSFRGAN and GPEN, inferring one image of 512x512 within 0.1s. Meanwhile, our method achieves the best performance in terms of LPIPS on the Celeb-Test dataset.
>
> **[Q3] Why are Stage II and III trained separately? How about training these two stages together?**
>
> It is necessary to train Stage II and III separately, since the settings of controllable coefficient *w* are different in these two training stages, i.e., *w* = 0 in Stage II, and *w* = 1 in Stage III. We tried to train these two stages together in an alternate iterative fashion, but cannot get a good performance. The separate training could greatly reduce the difficulty of network learning due to the same coefficient *w* at each stage.
>
> **[Q4] Provide discussions on the choosing of compression rate.**
>
> We set the compression ratio of 32 in our network design. Smaller compression ratios, such as 16, will destroy robustness to large degradation and result in a longer code sequence (32x32 for the compression ratio of 16), which will significantly increase the difficulty of global modeling of the Transformer and reduce its efficiency. Larger compression ratios, such as 64, cannot maintain the good fidelity of reconstruction results, leading to a serious issue of identity inconsistency. We will add this discussion in the revised version.
>
> **[Q5] Minor typos.**
>
> Thanks for your careful reading. We have fixed the typos in the updated manuscript.

---

> > ### Comment · Reviewer_ozVV · 2022-08-07
> > **The authors have addressed all my concerns and questions**
> >
> > The authors have addressed all my concerns and questions.

---

> > > ### Author Response · Authors · 2022-08-08
> > > **Thanks for your final rating**
> > >
> > > Thank you for giving the final rating as 'Accept'. We are glad that our answer addressed your concerns. Thanks for your time and valuable comments.

---

### Official Review · Reviewer_pfoY · 2022-07-06

**Rating:** 7
**Confidence:** 5
**Ethics Flag:** Yes
**Soundness:** 3 good
**Presentation:** 3 good
**Contribution:** 3 good

**Summary:**

To address the task of blind face restoration, this paper designs a novel network by casting restoration task to the code token prediction task. The authors propose a small and finite codebook space to reduce the uncertainty of image restoration. Upon the learned codebook, they exploit the global interaction by a Transformer, and the controllable modules are employed to balance quality and fidelity. The proposed method is evaluated on both synthetic and real-world datasets and extended to other tasks. The experimental results and ablation studies show that the proposed method is able to restore higher-quality results.

**Questions:**

*Questions
1. The training settings of Stage III is unclear, such as, what is $w$ value during training or is it a random value between 0 and 1?

2. The proposed method involves a Transformer to predict the code sequence, and the authors claim that code prediction task by the Transformer could ease the restoration task. But it is unclear if the advantages are introduced by the powerful Transformer. To demonstrate this statement, it would be better to conduct an ablation study that the Transformer predicts the features $\hat{Z}_c$ directly.
3. This method is interesting, but I wonder if it has a limitation when applied to generic natural images? From my side, it could be very challenging to learn an essential codebook for a natural image due to diverse scenes and complex textures. What about the authors’ ideas about this?

**Limitations:**

Yes

**Strengths And Weaknesses:**

*Strengths
1. The design philosophy of this work is well motivated: building a small discrete codebook to reduce the uncertainty of restoration; treating image restoration as a code prediction task; and adopting global-modeling transformer for better prediction.
2. The writing of this work is pretty good and it is easy to follow how the highlighted contributions are incorporated into the overall network.
3. The qualitative results are extensive and promising, especially in the provided supplementary and video demo. The proposed method is also evaluated on other tasks, including face inpainting, color enhancement, and old film enhancement.
4. I like the controllable design to make a trade-off between quality and fidelity, which is inspiring. A good balance between content preservation and generalization ability is achieved.
*Weaknesses
In general, the paper provides promising performance and sufficient experiments. There is no big flaw. Here, just I just provide some suggestions for improving the quality of the paper.

1. The main power seems to be driven by the learned codebook. In this paper, the authors utilize the codebook with 1024 items. Actually, the performance with the more codes in the codebook is also interesting though the 1024 seems sufficient for face image restoration. Discussion on the number of items could further complete this work.

2. Together with limitation discussion, some failure cases could be provided as well for better understanding.

---

> ### Author Response · Authors · 2022-08-02
> **Response to Assigned Reviewer pfoY**
>
> Thank you for the positive and insightful comments about the well-motivated idea, good writing, promising performance, extensive results, and inspiring design. We also appreciate the comments for providing further insightful discussions. Please check out the answers and discussions below.
>
> **[Q1] Further discussion on the number of code items in codebook.**
>
> We provided an ablation study in Sec. A.1 of the supplementary. Table 1 in suppl. shows that reconstruction performance (LPIPS and PSNR) is better as more codebook items are activated and learned.  The CodeFormer adopts a codebook with 1024 items. As suggested by Reviewer pfoY, we also train a large codebook with 2048 items, but it only brings a slight improvement in terms of LPIPS and PSNR, as shown in the following table. The results indicate that 1024 codes are sufficient to encode facial details. We will add this result in the revised supplementary.
> | Num. of Codebook | N=384 | N=1024 (CodeFormer) | N=2048 |
> | :--------------: | :---: | :-----------------: | :----: |
> | LPIPS&darr;      | 0.202 |        0.175        | 0.172  |
> | PSNR&uarr;       | 22.59 |        23.41        | 23.46  |
>
>
> **[Q2] Provide examples of failure cases.**
>
> We provide some failure cases via the anonymous link: https://www.dropbox.com/sh/rqc9lve9h9b0a1s/AACgvj-FbY3Xp1NSaB5-C-URa?dl=0 (download the .pdf file for the best view). Although CodeFormer exhibits great robustness in most cases, when it comes to side faces, CodeFormer offers limited superiority to other methods and also cannot produce good results for highly corrupted side faces. This is expected because there are only few side faces in the FFHQ training dataset. As a result, the codebook is unable to learn sufficient codes for the side faces, leading to less effectiveness in reconstruction and restoration.
>
> We will add some failure cases in our revised supplementary. We hope these failure cases could give some inspiration for future works. For example, one possible research direction is face restoration under different poses (including side faces), which is meaningful and challenging.
>
> **[Q3] The setting of tradeoff weight *w* in Stage III is unclear.**
>
> Sorry for confusing you with this training setting. Inspired by network interpolation [a], we only set the *w* to 1 as a marginal value during the training of stage III (Note that stage II was learned under another marginal value of *w* = 0), which then allows network to achieve continuous transitions of results by adjusting *w* in [0, 1] during inference. We have added the clear description in the updated manuscript (Line191).
>
> > [a] Xintao Wang et al. Deep Network Interpolation for Continuous Imagery Effect Transition. In CVPR, 2019.
>
> **[Q4] How to demonstrate the effectiveness of code prediction?**
>
> To verify the superiority of code prediction for codebook lookup, we conduct the ablation study of two variants, i.e., nearest-neighbour (NN) matching and a CNN-based code prediction module (adopt a Linear layer for prediction following the encoder). The comparison of Exps. (b) and (c) in Table 3 of the manuscript indicates that adopting code prediction for codebook lookup is more effective than NN feature matching.
>
> **[Q5] Wondering if it is possible to extend the proposed method to natural image restoration.**
>
> This is a good open question. The potential of generalizing the proposed CodeFormer to natural image restoration is also our next investigation. However, a few unique challenges in natural image restoration need to be addressed: 1) An essential and generic codebook for diverse scenes should be learned. 2) More efficient global modeling should be introduced due to the high resolution of natural images. 3) Additional priors such as semantic segmentation map may be needed, as the complexity and variety of textures in natural images will make codebook lookup more difficult and ambiguous.

---

> > ### Comment · Area_Chair_DJod · 2022-08-03
> > **A question**
> >
> > I have a question regarding the generalization ability of the proposed Codeformer method, which is also related to “[Q5]  Wondering if it is possible to extend the proposed method to natural image restoration”. Since the codebook and decoder are all fixed after the training is finished, what if one inputs into the trained network a degraded face whose high-quality version is considerably different from the trained high-quality faces? It would be better if the authors can include for comparison a baseline called “oracle Nearest Neighbor”, which performs restoration by using the most similar high-quality training face with respect to the high-quality version of the degraded testing face.

---

> > > ### Author Response · Authors · 2022-08-07
> > > **Response to AC DJod**
> > >
> > > Dear AC DJod, thanks for your question and discussion.
> > >
> > > The proposed CodeFormer actually has a good generalization ability to the unseen faces, even for real-world low-quality (LQ) faces. This is because the learned codebook is able to reconstruct high-quality faces not included in the training dataset. Instead of 'memorizing' the whole face image, the codebook is to learn base code items to store the context-rich visual parts of faces, which are able to represent any HQ faces beyond the training dataset by different code compositions. Theoretically, the representation space of a 1024 codebook with code composition (16x16) could produce $256^{1024}$ different HQ faces, which is much larger than the scale of the training dataset FFHQ (70,000). With such a large representation space, our method exhibits great expressive power and generalizability. In other words, we would like to say that our CodeFormer works by learning to predict the code combinations (sequences) of LQ faces, which are enormous and flexible representations, not 'retrieving' the HQ faces memorized from the training dataset. We list some results to support this statement as follows:
> > >
> > > 1. In our submission, the HQ versions of all evaluated faces in synthetic and real-world data are not included in the training dataset. (Please note that the real-world test dataset as well as old photos and movie clips do not have the corresponding HQ versions at all.) Nevertheless, the proposed CodeFormer still produced high-quality outputs.
> > >
> > > 2. To demonstrate the good generalizability of the learned codebook, we provide additional reconstruction results on some 'unseen' HQ faces from the CelebA-HQ dataset. Please find the results in `Response_to_AC_DJod.pdf`  (download for the best view) from this anonymous link: https://www.dropbox.com/sh/rqc9lve9h9b0a1s/AACgvj-FbY3Xp1NSaB5-C-URa?dl=0. As shown in Figure A, the learned codebook can reconstruct input HQ face images almost perfectly, even if they are not present in the training data.
> > >
> > > 3. As suggested by AC DJod, we evaluate the ”oracle Nearest Neighbor“ (oracle NN) on both synthetic LQs and real-world LQs (Note that there is no HQ version for real-world LQ). For each of the test LQ faces, we find the closest HQ face from the training dataset FFHQ in terms of Euclidean distance in the feature domain. We evaluate the oracle NN with different query images for NN retrieval as shown in the following table. The comparisons can be found in `Response_to_AC_DJod.pdf` (download for the best view) from this anonymous link: https://www.dropbox.com/sh/rqc9lve9h9b0a1s/AACgvj-FbY3Xp1NSaB5-C-URa?dl=0. From Figure B (synthetic ) and Figure C (real-world), we can see that oracle NN fails to retrieve a reasonable and matched HQ face, and its results show obvious identity inconsistency. This is expected because the training dataset FFHQ with 70,000 samples is far from being able to cover all faces, in spite of the fact that the FFHQ is the largest HQ face dataset in this field. Hence such a shortcut solution of oracle NN by retrieving faces from the FFHQ is not effective and cannot be applied in practice. Nevertheless, we agree such a comparison and discussion could gain more insight into the characteristics of this method.
> > >
> > >    | Method  | oracle NN (LQ) | oracle NN (HQ)|
> > >    | :-------- | :--- | :----------------- |
> > >    |  Query for synthetic data| synthetic LQ face |    corresponding HQ face (GT)    |
> > >    |  Query for real-world data| real-world LQ face |    N/A    |
> > >
> > >
> > > We hope this answer could solve your concerns well. Please feel free to give more comments for any concerns or questions. Thanks.

---

> > > > ### Comment · Area_Chair_DJod · 2022-08-07
> > > > **quantitative evaluation results for oracle NN**
> > > >
> > > > Thanks. The explanations are helpful. It would be better to show the quantitative evaluation results of oracle NN (HQ) in Table 1. This is purely for examining the diversity between training and testing data, not really a competitive comparision.

---

> > > > > ### Author Response · Authors · 2022-08-08
> > > > > **Quantitative evaluation results for oracle NN**
> > > > >
> > > > > Dear AC Djod, thanks for your valuable comments and suggestions. We evaluated the quantitative results of oracle NN (HQ) on the CelebA-Test dataset, as shown in the following table:
> > > > >
> > > > > | Methods | LPIPS&darr; |  FID&darr; | NIQE&darr; | IDS&uarr; | PSNR&uarr; | SSIM&uarr; |
> > > > > | :----- | :-----|:------ |:------ |:------- |:------- |:------ |
> > > > > | Input | 0.712 | 295.73| 18.67 |0.32 | 21.53 | 0.623 |
> > > > > | CodeFormer(ours) | 0.307 | 57.01 | 4.47 | 0.59 | 21.82 | 0.612 |
> > > > > | oracle NN (HQ) | 0.545 | 50.23 | 3.82 | 0.14 | 13.45 | 0.464 |
> > > > > |  Code GT | 0.124 | 54.31 | 4.33 | 0.89 | 25.43 | 0.749 |
> > > > > |  GT | 0.00 | 51.40 | 3.84 | 1.00 | ∞ | 1.00 |
> > > > >
> > > > > In terms of LPIPS, PSNR, SSIM, and IDS scores, oracle NN (HQ) shows poor quantitative performance, indicating a large disparity between training and testing data, which further demonstrates the good generalization ability of our method.
> > > > >
> > > > > We will add both quantitative and qualitative results of oracle NN in the final revised paper, and provide a discussion on the generalization ability of codebook to 'unseen' images.

---

> > ### Comment · Reviewer_pfoY · 2022-08-09
> > **The authors solve my concerns well. I would like to recommend acceptance.**
> >
> > The authors solve my concerns well. I would like to recommend acceptance.

---

### Official Review · Reviewer_BvZ9 · 2022-07-09

**Rating:** 5
**Confidence:** 1
**Soundness:** 2 fair
**Presentation:** 2 fair
**Contribution:** 2 fair

**Summary:**

This paper proposes a transformer-based blind face restoration method. Specifically, it follows VQ-GAN to compress the input features and find the nearest code, and performs autoregression and decoder to generate the synthesized images. To handle corrupted input images, they propose to fix the decoder and introduce a code prediction to ensure output fidelity. Extensive evaluations demonstrate the superior performance of the proposed method over existing approaches.

**Questions:**

- The proposed architecture follows VQ-GAN, except for the code prediction and controllable feature transformation. I am wondering if the VQ-GAN itself can also achieve similar restoration results.

**Limitations:**

I do not see any failure cases in the paper. It is essential to show some failure cases for the reader to investigate the failure model to benefit future research.

**Strengths And Weaknesses:**

Strengths
- The quantitative and visual results are significantly better than the existing methods.

Weaknesses
- This paper highly follows the architectural design of VQ-GAN. Although with extensive ablation studies, I am wondering if the VQ-GAN can also achieve similar restoration quality.

---

> ### Author Response · Authors · 2022-08-02
> **Response to Assigned Reviewer BvZ9**
>
> We thank the reviewer for the positive comments on the performance of the proposed method. We will answer your questions as below.
>
> **[Q1] Wondering if VQGAN itself can achieve similar restoration results, except for the code prediction and controllable feature transformation.**
>
> The original VQGAN performs codebook lookup via Nearest-Neighbor (NN) feature matching. Though it works well on the HQ features, it is not reliable for image restoration since the intrinsic textures of LQ inputs are usually corrupted. The information loss and diverse degradation in LQ images inevitably distort the feature distribution, prohibiting accurate feature matching. As depicted in Fig. 1(b)(right), even after fine-tuning the encoder on LQ images, the LQ features cannot cluster well as the HQ features do, but spread into other nearby code clusters. Hence, VQGAN, which adopts NN matching, is unreliable in such cases. This discovery is also the key motivation of this work.
>
> In the manuscript, we provided both qualitative and quantitative comparisons to demonstrate the necessity of our method and its advantages over VQGAN. **1)** The qualitative comparisons between VQGAN (NN) and CodeFormer in Fig. 1(f, g), Fig. 6, and Fig.8 demonstrate the superiority of our solution over the original VQGAN, producing much better results on both face restoration and inpainting. **2)** The quantitative comparison of VQGAN (NN) and CodeFormer provided in Table 3 (Exp. b and e) indicates that our method is more effective than NN-matching in VQGAN. In addition, the code lookup accuracy curves presented in Fig. 5 show that CodeFormer achieves more robust code prediction under different degradation levels.  **3)** The comparison of Exp. b and f in Table 3 shows that VQGAN produces lower fidelity results (IDS) due to the limited expressiveness of the codebook. Our CodeFormer offers controllable feature transformation modules that complement code composition errors and expression defects.
>
> Overall, the VQGAN (NN) itself cannot achieve similar restoration results as the proposed CodeFormer. In the case of heavy degradation, VQGAN usually produces low-quality results that are accompanied by artifacts due to inaccurate code lookup.  In addition, there is a serious issue of inconsistent identity in the outputs of VQGAN, which tends to produce lower fidelity results without any connection between encoder and decoder.
>
> **[Q2] Provide examples of failure cases.**
>
> We provide some failure cases via the anonymous link: https://www.dropbox.com/sh/rqc9lve9h9b0a1s/AACgvj-FbY3Xp1NSaB5-C-URa?dl=0 (download the .pdf file for the best view). Although CodeFormer exhibits great robustness in most cases, when it comes to side faces, CodeFormer offers limited superiority to other methods and also cannot produce good results for highly corrupted side faces. This is expected because there are only few side faces in the FFHQ training dataset. As a result, the codebook is unable to learn sufficient codes for the side faces, leading to less effectiveness in reconstruction and restoration.
>
> We will add some failure cases in our revised supplementary. We hope these failure cases could give some inspiration for future works. For example, one possible research direction is face restoration under different poses (including side faces), which is meaningful and challenging.

---

### Official Review · Reviewer_kois · 2022-07-10

**Rating:** 6
**Confidence:** 4
**Soundness:** 4 excellent
**Presentation:** 4 excellent
**Contribution:** 3 good

**Summary:**

This paper presents a vector quantized codebook based method for blind face restoration. Specifically, the face codebook is first learnt from an encoder-decoder convolutional network. To leverage global context, a transformer-structure neural network is designed to better predict code sequence for highly-corrupted LQ faces. At last, a controllable feature transformation module is trained to adjust the relative importance of the input face (fidelity). The proposed method achieves SoTA performance on synthetic dataset CelebA-Test and comparable performance on real-world datasets LFW-Test, WebPhoto-Test, and WIDER-Test. The ablation study also demonstrates the effectiveness of the proposed modules.

**Questions:**

My concerns and questions are:

1. I suggest providing sufficient discussion with some recent literatures, including:

[1] Yuchao Gu et al. VQFR: Blind Face Restoration with Vector-Quantized Dictionary and Parallel Decoder. arXiv preprint arXiv:2205.06803 (2022).
[2] Yang Zhao et al. Rethinking Deep Face Restoration. In CVPR, 2022.

2. I suggest providing detailed description on the network structure of the encoder and illustrate the design principle.

**Limitations:**

The limitations of the proposed method have been discussed in the submitted manuscript. I suggest providing quantitative examples for cases where the proposed method fails.

**Strengths And Weaknesses:**

Strengths:
1. Solid technical design and framework. The proposed CodeFormer framework is properly and elegantly designed, with supportive ablation studies. Remarkably, the proposed method is able to explicitly control and balance fidelity and quality.
2. Impressive qualitative performances shown in Figure 3, Figure 4, and the supplementary video. State-of-the-art or comparable quantitative performances shown in Table 1 and Table 2.
3. Clear writing style. Good technical presentation.

Concerns:
1. Missing related references and disscusion. I suggest to compare and discuss with some recent codebook-based works, e.g., [1] [2].
2. The network structure of the encoder in CodeFormer is unclear. The authors should give detailed description and illustrate the design insight.
3. Writing typos. (1) L144: Unexpected "?". (2) L169, Eq (6). Z_h should be Z_l.

[1] Yuchao Gu et al. VQFR: Blind Face Restoration with Vector-Quantized Dictionary and Parallel Decoder. arXiv preprint arXiv:2205.06803 (2022).
[2] Yang Zhao et al. Rethinking Deep Face Restoration. In CVPR, 2022.

---

> ### Author Response · Authors · 2022-08-02
> **Response to Assigned Reviewer kois**
>
> We appreciate the positive and constructive comments about the solid framework, elegant design, ablation study, impressive performance, and clear writing. The raised concerns are addressed as follows.
>
> **[Q1] Missing references and discussion with recent literature [1, 2].**
>
> **Discussion:** There are three key differences between the proposed CodeFormer and the two papers [1, 2]. **1)** The studies [1,2] perform codebook lookup using Nearest-Neighbour (NN) feature matching, which is not reliable for face restoration since the intrinsic textures of LQ inputs are usually corrupted. The information loss in LQ images inevitably distorts accurate feature matching, as depicted in Fig. 1(b) in the main paper. In contrast, our proposed CodeFormer performs codebook lookup via code prediction using a global-modeling Transformer, which shows superior robustness to degradation. The visual comparisons shown in Fig. 1(f, g), Fig. 6, and Fig. 8 support this statement. Besides, the code lookup accuracy comparison in Fig. 5 also demonstrates the superiority of our solution. **2)** Different from the works [1, 2] that fine-tune the decoder $D_H$ in their restoration training stage, we fix the decoder to protect the codebook prior from corruption. Our method emphasizes that the codebook must be used alongside the pre-trained decoder to fully unleash its potential. The ablation study of Exps. (e, g) in Table 3 shows that fine-tuning decoder deteriorates the performance, validating our statement. This is because fine-tuning the decoder destroys the learned prior held by the pre-trained codebook and decoder, resulting in suboptimal performance. **3)** Unlike the fixed fusion connections used in both studies [1, 2], we propose a controllable feature transformation module with an adjustable coefficient to control the information flow from the LQ encoder to decoder. Such a design allows a flexible trade-off between restoration quality and fidelity, achieving robustness against heavy degradation and good identity preservation within a single model. We will add a brief discussion in the revised version and add references [1, 2].
>
> **Comparison:** As the code of work [2] has not been released yet, we cannot make a comparison with it now. We have provided examples of visual comparisons between CodeFormer and VQFR [1]. Please find the comparison results from this anonymous link: https://www.dropbox.com/sh/rqc9lve9h9b0a1s/AACgvj-FbY3Xp1NSaB5-C-URa?dl=0 (download the .pdf file for the best view). The proposed CodeFormer produces better results than VQFR [1] on both heavy and medium degradation.
>
> > [1] Yuchao Gu et al. VQFR: Blind Face Restoration with Vector-Quantized Dictionary and Parallel Decoder. arXiv preprint arXiv:2205.06803 (2022).
> >
> > [2] Yang Zhao et al. Rethinking Deep Face Restoration. In CVPR, 2022.
>
>
> **[Q2] Provide the detailed network structure of the encoder in CodeFormer and illustrate the design insight.**
>
> We provide the detailed configurations of network structure in the anonymous link: https://www.dropbox.com/sh/rqc9lve9h9b0a1s/AACgvj-FbY3Xp1NSaB5-C-URa?dl=0, including the structure tables of Encoder, Decoder, Transformer module, and Controllable Feature Transformation module. As described in `Codebook Settings` (Sec. 3.1 in the manuscript), the Encoder mainly consists of 12 residual blocks and 5 downsampling conv layers.
>
> There is no special layer in Encoder, but we would like to emphasize that the compression ratio of 32 (5 downsampling layers) is critical in our design. Smaller compression ratios such as 16 will destroy robustness to large degradation and result in a longer code sequence (32x32 for the compression ratio of 16), which will significantly increase the difficulty of global modeling of the Transformer and reduce its efficiency. Larger compression ratios such as 64 cannot maintain good fidelity of reconstruction results, leading to a serious issue of identity inconsistency. We will also release our code and models for reference.
>
> **[Q3] Provide examples of failure cases.**
>
> We provide some failure cases via the anonymous link: https://www.dropbox.com/sh/rqc9lve9h9b0a1s/AACgvj-FbY3Xp1NSaB5-C-URa?dl=0. Although CodeFormer exhibits great robustness in most cases, when it comes to side faces, CodeFormer offers limited superiority to other methods and also cannot produce good results for highly corrupted side faces. This is expected because there are only few side faces in the FFHQ training dataset. As a result, the codebook cannot learn sufficient codes for the side faces, leading to less effectiveness in reconstruction and restoration.
>
> We will add some failure cases in our revised supplementary. We hope these failure cases could give some inspiration for future works. For example, one possible research direction is face restoration under different poses, which is meaningful and challenging.
>
> **[Q4] Writing typos.**
>
> Thanks for your careful reading. We have fixed the typos in the updated manuscript.

---

### Review · Ethics_Reviewer_xyvx · 2022-08-14

**Recommendation:**

I would suggest that the authors acknowledge the need for a bias analysis of the method e.g. how it performs on facial images of different color tones. The example of such an analysis is in this paper https://arxiv.org/abs/2003.03808 in Section 6 which was critiqued for similar reasons and which addressed the bias concerns. In the interest of clarifying potential negative impacts of the work, the authors should discuss the negative uses of their algorithm for example to de-anonymize images to violate privacy or to assist criminal investigations without proper scrutiny of the algorithm's performance. The paper shows face images from publically available dataset. However, it is not clear if the contributed images can be shown in a paper. It is not discussed if the data contributors provided consent to use their images. After including a discussion of potential negative impacts and clarifying the legitimate use of face images, the paper seems ok.

**Ethical Issues:**

Yes

**Ethics Review:**

The paper is flagged potentially because of bias/discrimination concerns with the output of the algorithm which restores blurred images. There is no discussion in the paper or in the rebuttal on bias properties of the proposed algorithm.

---

### Review · Ethics_Reviewer_5Qdb · 2022-08-18

**Recommendation:** No ethical issues

**Ethics Review:**

No ethical issues

---

### Author Response · Authors · 2022-08-09
**Update the Nonymous Link**

Dear reviewers and ACs,

We noticed that the server of the previous anonymized link (https://anonymous.4open.science/r/ID688/) is down due to unexpected errors. Thus, we replace it with the new anonymous link: https://www.dropbox.com/sh/rqc9lve9h9b0a1s/AACgvj-FbY3Xp1NSaB5-C-URa?dl=0

We have updated the link in each following response. Sorry for the inconvenience this may cause.

Thanks!

---

### Meta-Review · Area_Chair_DJod · 2022-08-20

**Recommendation:** Accept
**Confidence:** Certain

**Metareview:**

This work establishes a face restoration algorithm via integrating and optimizing several existing techniques, including VQ-GAN, Codebook prediction and Transformer. The key innovation comes from a Transformer-based prediction network, named CodeFormer, which may somehow exploit the global contexts helpful for codebook lookup. The experiments are reasonably designed, and the results are convincing. All the reviews agree that the paper is well-written and contains solid contributions, thus I would recommend accepting the paper.

**Award:**

No

---

### Decision · Program_Chairs · 2022-09-14

Accept